# Machine Learning Based Identification of Microseismic Signals Using Characteristic Parameters

**DOI:** 10.3390/s21216967

**Published:** 2021-10-20

**Authors:** Kang Peng, Zheng Tang, Longjun Dong, Daoyuan Sun

**Affiliations:** School of Resources and Safety Engineering, Central South University, Changsha 410083, China; pengkang@cqu.edu.cn (K.P.); tzheng@csu.edu.cn (Z.T.); lj.dong@csu.edu.cn (L.D.)

**Keywords:** microseismic monitoring, machine learning, source parameters, signal identification

## Abstract

Microseismic monitoring system is one of the effective means to monitor ground stress in deep mines. The accuracy and speed of microseismic signal identification directly affect the stability analysis in rock engineering. At present, manual identification, which heavily relies on manual experience, is widely used to classify microseismic events and blasts in the mines. To realize intelligent and accurate identification of microseismic events and blasts, a microseismic signal identification system based on machine learning was established in this work. The discrimination of microseismic events and blasts was established based on the machine learning framework. The microseismic monitoring data was used to optimize the parameters and validate the classification methods. Subsequently, ten machine learning algorithms were used as the preliminary algorithms of the learning layer, including the Decision Tree, Random Forest, Logistic Regression, SVM, KNN, GBDT, Naive Bayes, Bagging, AdaBoost, and MLP. Then, training set and test set, accounting for 50% of each data set, were prospectively examined, and the ACC, PPV, SEN, NPV, SPE, FAR and ROC curves were used as evaluation indexes. Finally, the performances of these machine learning algorithms in microseismic signal identification were evaluated with cross-validation methods. The results showed that the Logistic Regression classifier had the best performance in parameter identification, and the accuracy of cross-validation can reach more than 0.95. Random Forest, Decision Tree, and Naive Bayes also performed well in this data set. There were some differences in the accuracy of different classifiers in the training set, test set, and all data sets. To improve the accuracy of signal identification, the database of microseismic events and blasts should be expanded, to avoid the inaccurate data distribution caused by the small training set. Artificial intelligence identification methods, including Random Forest, Logistic Regression, Decision Tree, Naive Bayes, and AdaBoost algorithms, were applied to signal identification of the microseismic monitoring system in mines, and the identification results were consistent with the actual situation. In this way, the confusion caused by manual classification between microseismic events and blasts based on the characteristics of waveform signals is solved, and the required source parameters are easily obtained, which can ensure the accuracy and timeliness of microseismic events and blasts identification.

## 1. Introduction

Since the 1980s, with the gradual deepening of mining depth, mining accidents occur frequently in deep mines. The stress state of deep rock mass is significantly different from that of shallow rock mass in engineering because the deep rock mass is in a special environment of high stress, high temperature, high permeability, and strong mining disturbance [1]. The process of micro-crack expansion in the rock mass is difficult to be captured by traditional rock mass stability monitoring technology. When a measurable displacement occurs on the surface of the rock mass, a considerable fracture or sliding may have occurred inside the rock mass. After years of practice and improvement, microseismic monitoring technology has become one of the effective means of stability monitoring for the deep mine. In the mining process, elastic deformation and inelastic deformation are generated in the rock mass. During the inelastic deformation process, the elastic potential energy accumulated inside the rock mass will be released in the form of a vibration wave [1,2], which will be received by the sensor arranged by the microseismic monitoring system and defined as a microseismic event. A large number of microseismic events will occur in the process of rock deformation and internal crack propagation. The essence of microseismic events is the manifestation of a series of dynamic processes, such as stress-strain and instability-failure of rock mass [2]. Microseismic events contain abundant source information: microseismic events received by sensors around the source have been used for seismic source localization [3,4,5] and abnormal region identification [6,7]; the waveform of microseismic events was received to inverse the source mechanism [8,9,10,11]; microseismic events were used to predict rock burst and other rock disasters and provide reliable data support for regional stability analysis of rock masses [12,13,14,15]. For different research objects, the events collected by the microseismic monitoring system are classified into two categories: (1) microseismic events generated in the rock deformation and micro-crack expansion inside the rock mass [16]; and (2) blasts of rock mass caused by the impact wave directly generated by the underground dynamite explosion [17,18].

The analysis of microseismic events is based on accurate and pure microseismic monitoring signals, which requires eliminating the interference signals such as blasting and noise before analysis. Although the microseismic monitoring system is maturely used to analyze the stability of rock mass, it is difficult to directly extract accurate microseismic information in a complex environment, especially under the interference of various noise and explosions. Traditional methods for identifying the microseismic events and blasts depend on the manual or subjective experience, resulting in large errors in discrimination and parameter analysis. The research on identification of controlled blasts and natural microseismicities is mainly divided into the differences in frequency distributions, source parameters, and waveforms. Statistical analysis and spectrum analysis, as well as machine learning, are the three kinds of widely used methods in current research.

Some researchers adopted the Fourier transform to obtain the difference between microseismic events and blasts, which provided a basis for the classification of microseismic events and blasts [19,20]. Frantti and Levereault [21] converted the frequency of the seismic and blasting signals into an available range through hundred times of compression and correctly reclassified 2/3 seismic signals. Tayler [22] proposed a high-frequency *P_g_*/*L_g_* discriminant about earthquakes and blasts between 0.5 and 10 Hz. As the increase of frequency, the dividing line between the two signals became obvious. The identification accuracy reaches 95%. Jiang et al. [23] classified the microseismic and blasting signals based on the fast Fourier transform spectrum analysis. By comparing the spectrums of the two kinds of signals, they found that the blast signals have the characteristics of high energy release and shock reaction time. The energy of blast signals was mostly distributed in the region of 0–30 Hz. On the contrary, the energy of the microseismic signals was mainly distributed in the frequency range of 30–50 Hz. Zhao et al. [24] used a frequency slice wavelet transform (FSWT) to study the typical microseismic and blast vibration signals of the rock mass in a mine. The results showed that the energy of the two kinds of signals is mainly distributed below 100 Hz and the energy of rock mass microseismic signals was mainly concentrated in the band between 0~50 Hz, while the energy of the blast vibration signals is concentrated more obviously in the band between 50~100 Hz. In addition, blasting vibration signals accounted for a larger proportion of energy in areas above 100 Hz. He et al. [25] used the Mel-frequency cepstral coefficients method to convert four types of signals (rock burst, blasting vibration, electromagnetic interference, and rig drilling) into the nonlinear frequency spectrum on the MEL scale and then switch to the cepstrum domain. Combining with the difference in the time domain, they chose a set of 24 dimensions as a characteristic parameter vector, which is used to construct and train the Gaussian mixture hidden Markov identification model. The accuracy of the classification of microseismic events reached 92.46%. Spectrum analysis is hard to be applied for classifying the blast and microseismic signals in the field operation on account of a large number of sensors in the microseismic monitoring system and the high requirement of spectrum analysis on the professional knowledge, although it is a relatively accurate method.

Energy will be released in the type of microseismic waves by cracks in rocks. The source parameters of microseismic events will be different depending on the fracture mode of rocks. Based on the above characteristics, the characteristic parameters of the received microseismic events can be used as a criterion to distinguish the two kinds of signals. Muller et al. [26] classified mining blasts and rock bursts according to the earthquake seismometer network. In their method, a multi-layer neural network was adopted for fusion, and fuzzy coding was conducted for the input characteristics of the neural network, combined with the characteristics of the signals collected by sensors. Orlic and Loncaric [27] proposed a new method to classify the natural and artificial earthquake signals, in which the near best seismic features were searched by a genetic algorithm. Although the source parameters can be regarded as the difference between the two kinds of signals, the source parameter analysis relies on the subjective parameter selection and judgment of researchers. Additionally, the parameters’ correlations are rarely considered when selecting the parameters. As a result, it will significantly affect the identification accuracy. The pre-analysis of the parameters is needed before the establishment of the model, and the applicability of different parameters to different classification models is unknown, thus the computational complexity is increased inevitably. The discrimination of blasts and microseismic events is mainly based on monitoring parameters or waveform contained in the events previously. With the quick and efficient development of modern artificial intelligence (AI), machine learning can also be applied to identify blasts and microseismic events. Dowla [28] applied a neural network to the classification of microseismic events. Shang et al. [29] analyzed the microseismic events and related parameters by PCA and ANN and found that the collaborative method of PCA and ANN achieved the best classification accuracy compared to other methods.

Spectral analysis is not suitable for the classification of the mine database because of its high requirement for professional knowledge. Statistical analysis methods often rely on manual experience to select the parameters and models, without considering the correlation between parameters. In addition, the classification of microseismic events and blasts has a serious lag. In this work, ten machine learning methods including Decision Tree, Random Forest, Logistic Regression, Support Vector Machine (SVM), Gradient Boosting Decision Tree (GBDT), K-nearest neighbor (KNN), AdaBoost, Naive Bayes, Bagging, and Multi-layer Perceptron classification (MLP) are used to establish models for classification of microseismic events and blasts by 100 collected cases. Under the condition of a small sample data set, ten different machine learning algorithms are used to find out the best training model and its optimal parameters.

## 2. Materials and Methods

### 2.1. Event Features

The energy generated by the microfracture in the rock mass will be released in the form of a seismic wave. Through the microseismic monitoring system, different kinds of source parameters can be collected. Due to the distinct source mechanism of microseismic events and blasts, their characteristic parameters are different [30].

The principle of the characteristic parameter selection for the classification of microseismic events and blasts is as follows: characteristic parameters from the same category should be as similar as possible, while characteristic values from different categories should be as distinct as possible. The data sets used in this work are all from the microseismic monitoring system in mines, which are shown in Appendix A.

By drawing a probability density histogram, parameters with a certain recognition effect can be found, as shown in Figure 1. Finally, the seismic moment M, energy E, the number of triggered sensors N, the first peak A, the time of maximum peak T, and the dominant frequency F are selected as the characteristic parameters to classify the two kinds of signals. Table 1 shows the statistical characteristics of the collected samples.

### 2.2. Methods

In this system, machine learning was used to classify and recognize microseismic events and blasts. The system was composed of data feature area, machine learning area, and model prediction area. Ten machine learning methods were used to classify microseismic events and blasts, including Decision Tree, Random Forest, Logistic Regression, SVM, KNN, Naive Bayes, GBDT, AdaBoost, Bagging, MLP, as shown in Figure 2. The training samples and test samples accounted for 50% of the data set, respectively. ACC, PPV, SEN, NPV, SPE, FAR and ROC curves were used as evaluation indexes to analyze the classification results.

Decision Tree can be expressed as a binary tree, each leaf node represents an input variable and a variable-based bifurcation. The prediction result of the Decision Tree is obtained as follows: (1) follow the tree’s bifurcation path until it reaches a leaf node and (2) output the category of the node.

Based on Bagging, the randomness is introduced in the Random Forest and to split the data sets. In this way, the model created for each sample is more random with higher accuracy than that created in other cases, and output results can be more consistent with the real value.

Logistic Regression is a classification method derived from statistics, the purpose of which is to find the weight value of each input variable, and the final output prediction result is obtained through a nonlinear function transformation.

SVM aims to find a hyperplane that divides the input variable space into categories. All the input points can be completely divided by the hyperplane. The training process of SVM is actually to search for the coefficients of the optimal category segmentation.

In KNN algorithm, the classification result is obtained as follows: (1) search the whole training set; (2) find out the most similar K instances with a certain data point; (3) summarize the output variables of these K instances; and (4) output the category of the data point. For the regression problem, the prediction result is the mean value of the output variables, while for the classification problem, the prediction result is a certain category.

Naive Bayes algorithm uses two kinds of probabilities (prior probability and posterior probability) which can be directly obtained from the training data to predict the new data. By assuming that each input variable is independent of each other, the algorithm has an outstanding performance in the calculation of complex problems.

Based on the boosting algorithm, GBDT adopts the idea of gradient lifting. In this algorithm, the sum of the results of all weak classifiers is equal to the predicted value, and then the next weak classifier is used to fit the error between the predicted value and the real value. The final result of this algorithm is determined by multiple trees as the Random Forest. The trees that make up the Random Forest can be either regression trees or decision trees, while GBDT is only composed of regression trees. In addition, the result of the Random Forest is determined by a majority of votes, and GBDT is determined by the accumulation of the results of multiple trees.

In the AdaBoost algorithm, the weight of the training samples in the latter tree is obtained based on the performance of the training samples in the previous tree. The weight of the unpredictable training samples increases in the latter tree, and the weight of the easily predicted training samples decreases in the latter tree. The performance of each tree affects the weight of training in the latter tree. Finally, the final results are weighted according to the accuracy of each tree in training.

In the Bagging algorithm, multiple samples in the training set are selected to build a model for these sample sets. When classifying the test set, the prediction result is generated by each model established before. Finally, the average value of all models is taken as the final classification result of the Bagging algorithm.

MLP includes an input layer, an output layer, and at least one hidden layer. In the hidden layer, the neurons in the upper layer are connected with all the neurons in the lower layer. In the training process of training samples, the weights and bias parameters in the hidden layer are adjusted continuously to make the output value consistent with the real value.

Then, 10 machine learning models were evaluated by cross-validation. In the process of cross-validation, K = 10 was used as the evaluation parameter, i.e., the whole data was randomly divided into K parts, and the K-1 part was used as the fit model of the training set, and the other part was used as the evaluation model of the test set. By repeating K times, the K models and performance evaluation results were obtained, and the average performance was calculated.

## 3. Results and Discussion

### 3.1. Assessment Results Using Traditional Models

The study on microseismic signal identification can be roughly divided into spectrum analysis, statistical analysis, and artificial intelligence method involved in this paper. According to the literature, spectrum transformation, neural network, and Fisher discrimination have been used for analysis in the past (Table 2). Among the results involved in the reference, the artificial intelligence method has achieved a more efficient identification performance. Due to the high demand for professional knowledge in the application, spectrum analysis is not widely used in the field. Although the Fisher discriminant analysis method has a good performance in the identification of the test set, this method depends on subjective experience in parameter selection and modeling and has certain randomness. With the development of artificial intelligence (AI), the AI identification method shows superiority such as simple, objective operation, and high accuracy in the identification of microseismic signals. At present, there are many artificial intelligence algorithms. However, in the previous literature, common issues were simply compared, and the corresponding results were not evaluated. In this paper, ten common machine learning classification algorithms are selected to perform the microseismic signal identification. Meanwhile, the identification results and parameters are analyzed and evaluated.

### 3.2. Assessment Results Using Ten Machine Learning Classifiers

Ten machine learning algorithms are used to classify microseismic events and blasts. The results show that except for Logistic Regression, KNN, Naive Bayes, and Bagging algorithms, the recognition accuracy of other algorithms to the training set is 100%; the performance of Random Forest and KNN on the test set is better, reaching 94%, and the accuracy of Logistic Regression and Bagging algorithm can achieve 92%. Although SVM algorithm shows excellent identification ability in the training set, its accuracy rate in the test set is only 68%. In addition, there is an obvious overfitting phenomenon. Hence, under the condition of small sample data set, SVM cannot meet our requirements for sample fitting. Table 3 and Table 4 show the accuracy rate and various evaluation indexes of ten algorithms in the training set and test set. As shown in the probability density distribution diagram, there are obvious differences between parameter lgA and parameter lgT in the training set and test set. Hence, the two parameters are selected to draw the decision boundary diagram (Figure 3 and Figure 4). The decision boundary diagram shows different decision boundaries in different methods. Through comparison, it is found that the decision boundaries of the integrated algorithm based on GBDT and Random Forest are clear, and these algorithms perform well in the training set and test set.

For the classification results of test set samples, ACC, PPV, SEN, NPV, and SPE indexes of Random Forest and KNN algorithm are all greater than 0.900; and for the Logistic Regression and Bagging, except that the PPV and SPE indexes are less than 0.900, the other indexes are also greater than 0.900. Therefore, in this sample condition, several machine learning algorithms, such as Random Forest, can effectively identify microseismic events and blasts according to source parameters.

### 3.3. Discussion

The evaluation results of cross-validation (K = 10) for ten machine learning algorithms are shown in Table 5. There is a difference between the results of cross-validation and the performance in the training set and test set, because the original data set is divided into ten parts in the cross-validation, nine of which are used as training set and one of which is used as the test set. After ten times of verification groups by the above method, the accuracy is averaged as the final result. Thus, the cross-validation can evaluate the performance of the model more accurately. Since the information content of the training sample is larger than that of the test sample, the trained model has a stronger identification ability. It suggests that a larger training data set can improve identification accuracy. Table 6 shows the evaluation results of ten machine learning algorithms through cross-validation and parameter settings in the case of the optimal model.

According to the accuracy of each machine learning algorithm after cross-validation, the performance of this method in the training set, test set, and all data set is scored. The score is visually expressed by the symbol:

***** indicates the algorithm with an accuracy greater than 0.95;

**** indicates the algorithm with an accuracy greater than 0.90 and less than 0.95;

*** indicates the algorithm with an accuracy greater than 0.85 and less than 0.90;

** indicates the algorithm with an accuracy greater than 0.80 and less than 0.85;

* indicates the algorithm with an accuracy less than 0.80.

According to the results of model evaluation, the performance of the Logistic Regression algorithm is the best in the three data sets, followed by Random Forest and Naive Bayes algorithm. Random Forest and Naive Bayes obtain ***** in the training set and all data set, and **** in the test set.

## 4. Conclusions

Based on ten common machine learning algorithms, microseismic events and blasts are classified using six source parameters collected by the microseismic monitoring system, namely seismic moment, energy, number of triggered sensors, first peak, time of the maximum peak, and dominant frequency. Compared with the traditional classification methods, the proposed method reduced the errors caused by the differences of human experience by using the machine learning framework. Through the classification of the monitoring data using the established models, it is found that the efficiency and accuracy of the signal identification were improved. In addition, by comparing the manual division of the training set and test set with cross-validation, it is found that the quality of training samples directly affects the recognition accuracy of the model. To improve the classification accuracy of microseismic events and blasts, parameter samples should be added continuously to enlarge the training data.

## Figures and Tables

**Figure 1 sensors-21-06967-f001:**
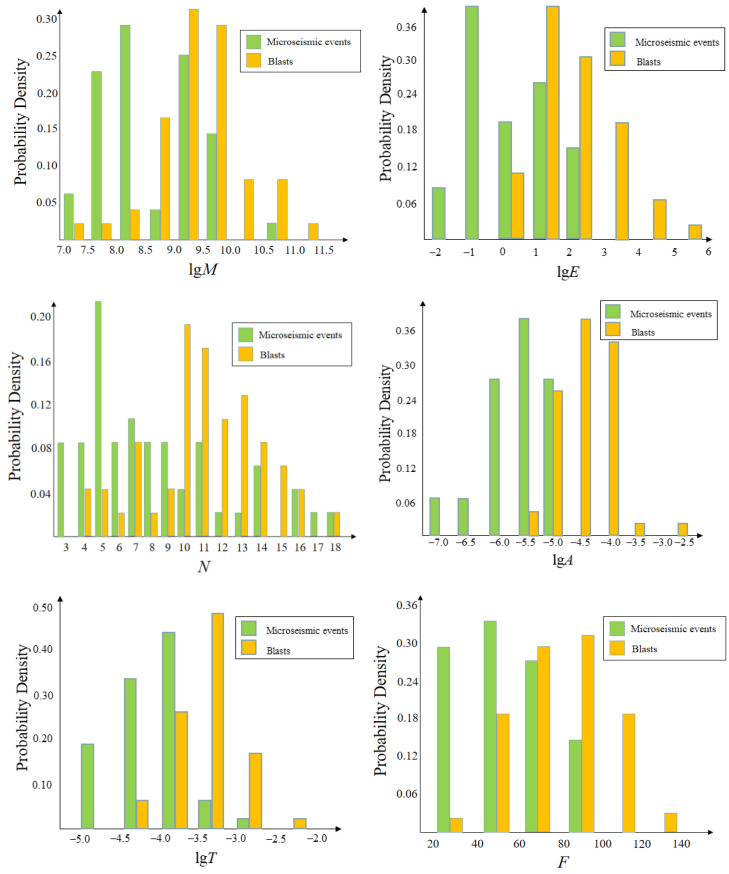
Probability density functions of characteristics.

**Figure 2 sensors-21-06967-f002:**
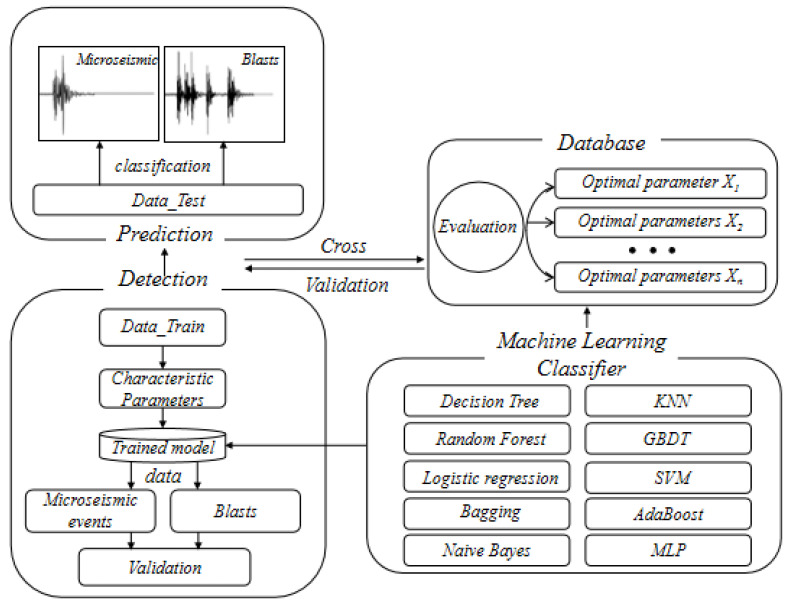
Classification of microseismic events and blasts using machine learning algorithms.

**Figure 3 sensors-21-06967-f003:**
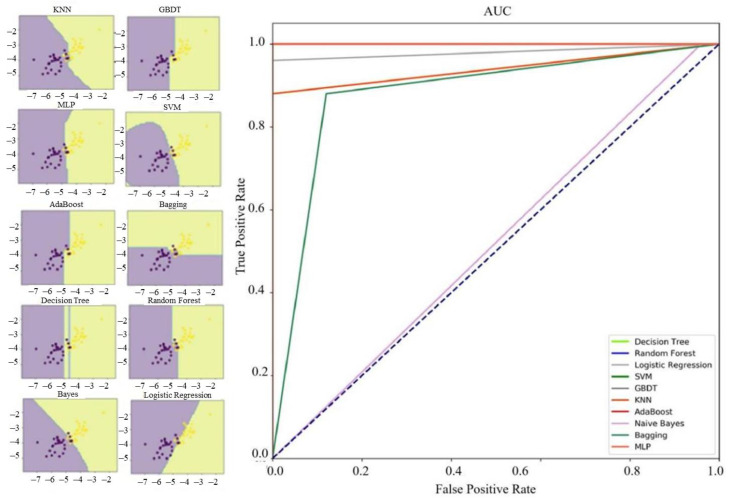
Decision boundary and ROC curve of ten methods for the training set.

**Figure 4 sensors-21-06967-f004:**
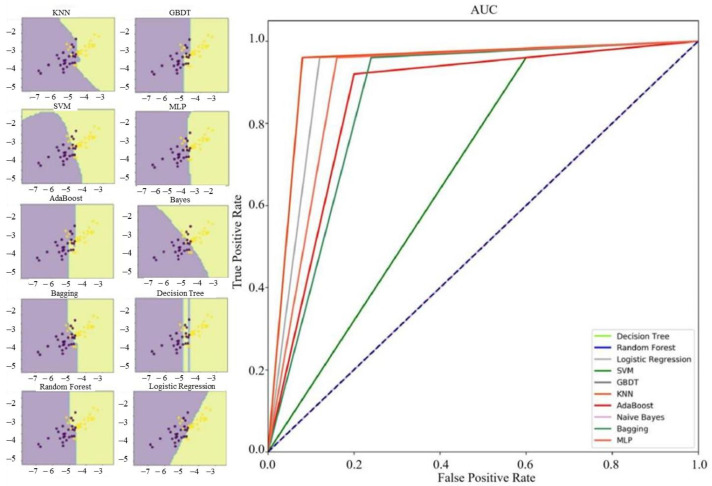
Decision boundary and ROC curve of ten methods for the test set.

**Table 1 sensors-21-06967-t001:** Characteristics of the samples.

Index	Data Set of Microseismic Events		Data Set of Blasts	
	Mean	SD	Mean	SD
*lgM*	8.6416	0.792632813	9.4202	0.762676517
*lgE*	0.5806	1.245097177	2.3396	1.140919715
*N*	8.06	3.991879512	10.9	3.183182851
*LgA*	−5.3976	0.580209533	−4.1588	0.521393893
*lgT*	−4.0314	0.452124283	−3.328	0.441546448
*F*	35.7	18.61629745	79.958	23.38982938

**Table 2 sensors-21-06967-t002:** Traditional classification model evaluation.

Scholar	Methods or Objects	Accuracy
Malovichko [18]	Multivariate maximum likelihood Gaussian classifier	20% reclassify
Frantti and Levereault [21]	Spectrum analysis	2/3
Tayler [22]	Maximum likelihood Gaussian + BP neural network	95%
Jiang et al. [23]	FFT spectrum analysis	
Zhao et al. [24]	Linear regression + Fisher discriminant	97.1%
Muller et al. [26]	Neural network	90%
Orlic and Loncaric [27]	Genetic algorithm	85%
Vallejos and McKinnon [17]	Logistic Regression and neural work	95%

**Table 3 sensors-21-06967-t003:** Evaluation results of machine learning model on training set.

Classifier	ACC	PPV	SEN	NPV	SPE	FAR
Decision Tree	1.000	1.000	1.000	1.000	1.000	0.000
Random Forest	1.000	1.000	1.000	1.000	1.000	0.000
Logistic Regression	0.980	1.000	0.962	0.960	1.000	0.000
SVM	1.000	1.000	1.000	1.000	1.000	0.000
GBDT	1.000	1.000	1.000	1.000	1.000	0.000
KNN	0.940	1.000	0.893	0.880	1.000	0.000
AdaBoost	1.000	1.000	1.000	1.000	1.000	0.000
Naive Bayes	0.980	1.000	0.962	0.960	1.000	0.000
Bagging	0.980	1.000	0.962	0.960	1.000	0.000
MLP	1.000	1.000	1.000	1.000	1.000	0.000

**Table 4 sensors-21-06967-t004:** Evaluation results of machine learning model on test set.

Classifier	ACC	PPV	SEN	NPV	SPE	FAR
Decision Tree	0.860	0.800	0.909	0.920	0.821	0.178
Random Forest	0.940	0.920	0.958	0.960	0.923	0.077
Logistic Regression	0.920	0.880	0.956	0.960	0.889	0.111
SVM	0.680	0.400	0.909	0.960	0.615	0.385
GBDT	0.860	0.800	0.909	0.920	0.821	0.179
KNN	0.940	0.920	0.958	0.960	0.923	0.077
AdaBoost	0.860	0.800	0.909	0.920	0.821	0.178
Naive Bayes	0.820	0.640	1.000	1.000	0.735	0.265
Bagging	0.920	0.840	1.000	1.000	0.862	0.138
MLP	0.900	0.840	0.955	0.960	0.857	0.143

**Table 5 sensors-21-06967-t005:** Optimal accuracy of the ten machine learning algorithms and parameter settings.

Group	Model	ACC	Parameters
All Data	Decision Tree	0.940	Criterion = ‘gini’ Max_depth = 3 Min_impurity_decrease = 0.0 Min_samples_leaf = 1 Splitter = ‘best’
	Random Forest	0.960	N_estimators = 31 Max_depth = 6 Min_samples_leaf = 1 Min_sanmples_split = 2 Criterion = ‘entropy’
	Logistic Regression	0.950	C = 1.0 Class_weight = ‘balanced’ Solver = ‘liblinear’
	SVM	0.870	Kernel = ‘rbf’ Probability = True
	GBDT	0.950	
	KNN	0.940	N_neighbors = 5
	AdaBoost	0.960	Max_depth = 2 Min_samples_split = 20 Min_samples_leaf = 5 Algorithm = ‘SAMME’ N_estimators = 200 Learning_rate = 0.8
	Naive Bayes	0.950	Priors = None Var_smoothing = 1 × 10^−9^
	Bagging	0.900	Max_samples = 0.5 Max_features = 0.5
	MLP	0.950	Solver = ‘lbfgs’ Alpha = 1e-5 Hidden_layer_sizes = (30,20) Random_state = 1
Training Data	Decision Tree	0.920	Criterion = ‘gini’ Max_depth = 3 Min_impurity_decrease = 0.0 Min_samples_leaf = 1 Splitter = ‘random’
	Random Forest	0.983	N_estimators = 11 Max_depth = 3 Min_samples_leaf = 1 Min_sanmples_split = 2 Criterion = ‘gini’
	Logistic Regression	0.980	C = 1.0 Class_weight = ‘balanced’ Solver = ‘liblinear’
	SVM	0.775	Kernel = ‘rbf’ Probability = True
	GBDT	0.891	
	KNN	0.892	N_neighbors = 2
	AdaBoost	0.967	Max_depth = 2 Min_samples_split = 20 Min_samples_leaf = 5 Algorithm = ‘SAMME’ N_estimators = 200 Learning_rate = 0.8
	Naive Bayes	0.975	Priors = None Var_smoothing = 1 × 10^−9^
	Bagging	0.875	Max_samples = 0.5 Max_features = 0.5
	MLP	0.958	Solver = ‘lbfgs’ Alpha = 1 × 10^−5^ Hidden_layer_sizes = (30,20) Random_state = 1
Test Data	Decision Tree	0.940	Criterion = ‘gini’ Max_depth = 2 Min_impurity_decrease = 0.0 Min_samples_leaf = 1 Splitter = ‘best’
	Random Forest	0.933	N_estimators = 91 Max_depth = 4 Min_samples_leaf = 1 Min_sanmples_split = 2 Criterion = ‘gini’
	Logistic Regression	0.960	C = 1.0 Class_weight = ‘balanced’ Solver = ‘liblinear’
	SVM	0.675	Kernel = ‘rbf’ Probability = True
	GBDT	0.833	
	KNN	0.858	N_neighbors = 4
	AdaBoost	0.850	Max_depth = 2 Min_samples_split = 20 Min_samples_leaf = 5 Algorithm = ‘SAMME’ N_estimators = 200 Learning_rate = 0.8
	Naive Bayes	0.900	Priors = None Var_smoothing = 1 × 10^−9^
	Bagging	0.833	Max_samples = 0.5 Max_features = 0.5
	MLP	0.892	Solver = ‘lbfgs’ Alpha = 1 × 10^−5^ Hidden_layer_sizes = (30,20) Random_state = 1

**Table 6 sensors-21-06967-t006:** Evaluation results of ten machine learning algorithms.

Methods	Training Data	Test Data	All Data
Decision Tree	****	****	****
Random Forest	*****	****	*****
Logistic Regression	*****	*****	*****
SVM	*	*	***
GBDT	***	**	*****
KNN	***	***	****
AdaBoost	*****	***	*****
Naive Bayes	*****	****	*****
Bagging	***	**	****
MLP	*****	***	*****

## Data Availability

Thanks you. The details of the data supporting our reported results are listed in Appendix A.

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
