# Peer review of "Machine Learning Based Identification of Microseismic Signals Using Characteristic Parameters"

_sensors, 2021, doi:10.3390/s21216967_

Round 1

Reviewer 1 Report

In the present work, the proposal consists of Microseismic Signals detection with machine learning algorithms. The manuscript contribution is an evaluation of ten different machine learning algorithms.

After reading the manuscript, I have some considerations described below.

1 - The English from the manuscript must be improved.

2 - The authors should make clear which are the contributions.

3 - I miss a description of the dataset. The number of samples per class, the necessary preprocessing, the input size used in the methods. Is it a private dataset?

4 - I was not able to understand the data division. In some points, the authors state "The training samples and test samples accounted for 50% of the data set, respectively.", in another point they state: "In the process of cross-validation, K = 10 was used as the evaluation parameter...". Which data distribution is used in each experiment?

5 - The authors report several metrics but they do not specify how to calculate them. They only give the acronyms for them. The authors should either define them or give the full name and a reference.

6 - I also miss a comparison with other works from the literature.

7 - Finally, are the authors planning to make public the source of the proposed approach? It could help future researchers and make easier a direct comparison in a different scenario. 

Thus, in my judgment, this manuscript should not be considered for publication. I believe that the study is not suitable for the Sensors Journal. 

Author Response

Dear Reviewer,

Thank you very much for the review of our manuscript, entitled “Identification Method of Microseismic Signals using Characteristic Parameters Based on Machine Learning”. We have revised the manuscript according to your comments and suggestions. For detailed revision, please see the attachment.

Attached with this reply are:

A list of responses to your comments; and

A revised version of the manuscript (All of the revisions are marked with yellow in the revised manuscript).

Kind regards,

Sincerely,

Daoyuan Sun,

School of Resources and Safety Engineering, Central South University

E-mail: sundaoyuan@csu.edu.cn

Reviewer 2 Report

The papers does not have much novelty in it. It just extracts 6 features from seismic signals and trains 10 popular (well known) machine learning models and compares the results. This  is usually done by an MSc student in the final project. The authors should increase the novelty of the paper.

Figure 2 is confusing:  Do you have only 3 parameters in the top-left Database box? I do not think so. And Accuracy box in the Detection Box should be deleted as it complicates the understanding more than it eases it.

I do not see any details about the dataset used. How big was it, how did you acquired the dataset? How challenging was it in terms of data processing? Was the dataset available somewhere for others to use? None of these details appear in the paper.

In conclusion the paper appears quite simple, with not much contribution to machine learning and also to the application area as well (for example you should talk more about the application, challenges etc etc. which should be OK if the paper is application oriented even if the machine learning content is basic)

Author Response

(The authors gave the same response as above.)

Reviewer 3 Report

Summary:

This paper presents a microseismic signal identification system based on machine learning. Based on the data of source parameter of the physical acquisition layer, ten different algorithms are used as the preliminary algorithms of learning layer and their performances in microseismic signal identification are evaluated by cross validation. To improve the accuracy of signal identification, the database of microseismic events and blasts should be expanded, so as to avoid the inaccurate data distribution caused by the small training set. It is shown that the confusion caused by manual classification between microseismic events and blasts based on the characteristics of waveform signals is solved using the required source parameters which can ensure the accuracy and timeliness of microseismic events and blasts identification.

Major comments:

- Overall, I find the paper is easy to follow and the experimental evaluation shows promising results, but my major concern is about the novelty of this work (evaluation of established methods).

- Very generic discussion of machine learning approaches.

- I would recommend the authors to provide more details on model training for different algorithms.

Minor comments:

- The axis labels in Figure 3 appear too small, which may hinder readability.

Author Response

(The authors gave the same response as above.)

Round 2

Reviewer 1 Report

After reading the paper, in my judgment, I still believe that this manuscript should not be considered for publication. The manuscript has some serious flaws in the construction of its methodology and experiments.

Author Response

(The authors gave the same response as above.)

Reviewer 2 Report

The paper has some good technical work and results but it is not written well, especially in the new added sentences. Many of these sentences are not professional from an ML point of view, they should be checked by someone working in ML. Some examples:

The title is not very good:  "Identification Method of Microseismic Signals using Characteristic Parameters Based on Machine Learning", what is based on ML, parameters or identification method?? It sounds bad in English. Better "Machine Learning based Identification of Microseismic Signals using Characteristic Parameters", although even this is not very specific but better.

Line 4-5 in Abstract "To remedy this situation, ...." it is not a defect. 

This sentence needs reformulation, it does not sound professional "Based on the source parameter data of the physical acquisition layer, the parameter fitting model was established using the machine learning layer, and the received data were identified in the classification layer. "

"were evaluated with cross-validation methods." not results

"The system was composed of data feature module, machine learning module, and model prediction module." what is the difference between Ml module and model prediction model?  (this is also an ML module). please check.

"Based on Bagging, the randomness is introduced in the Random Forest and to split the data sets" not "Based on Bagging, the randomness is introduced by establishing a decision tree inRandom Forest, to realize the sub-optimal segmentation of data sets. " or reformulate.

Pay attention to the new section in Conclusions which do not sound well

Author Response

(The authors gave the same response as above.)
